# TFTF: An R-Based Integrative Tool for Decoding Human Transcription Factor–Target Interactions

**DOI:** 10.3390/biom14070749

**Published:** 2024-06-24

**Authors:** Jin Wang

**Affiliations:** School of Public Health, Suzhou Medical College, Soochow University, Suzhou 215123, China; jinwang93@suda.edu.cn

**Keywords:** transcription factors, gene regulatory networks, bioinformatics, TF–target prediction, Shiny app

## Abstract

Transcription factors (TFs) are crucial in modulating gene expression and sculpting cellular and organismal phenotypes. The identification of TF–target gene interactions is pivotal for comprehending molecular pathways and disease etiologies but has been hindered by the demanding nature of traditional experimental approaches. This paper introduces a novel web application and package utilizing the R program, which predicts TF–target gene relationships and vice versa. Our application integrates the predictive power of various bioinformatic tools, leveraging their combined strengths to provide robust predictions. It merges databases for enhanced precision, incorporates gene expression correlation for accuracy, and employs pan-tissue correlation analysis for context-specific insights. The application also enables the integration of user data with established resources to analyze TF–target gene networks. Despite its current limitation to human data, it provides a platform to explore gene regulatory mechanisms comprehensively. This integrated, systematic approach offers researchers an invaluable tool for dissecting the complexities of gene regulation, with the potential for future expansions to include a broader range of species.

## 1. Introduction

Transcription factors (TFs) are pivotal regulatory proteins that modulate gene expression by binding to specific DNA sequences, thus orchestrating cellular function and organismal development. Unraveling the complex interactions between TFs and their target genes is crucial for understanding the molecular underpinnings of biological processes and disease states [1,2]. Traditionally, the identification of interactions between transcription factors (TFs) and their target genes has relied on labor-intensive and time-consuming experimental methods. However, with the rapid advancement of high-throughput analytical techniques, particularly chromatin immunoprecipitation sequencing (ChIP-seq) and RNA sequencing (RNA-seq), it has become possible to predict TF target genes on a genomic scale [3,4]. ChIP-seq maps the associations between TFs and DNA, while RNA-seq identifies changes in RNA levels associated with perturbations in TF activity [5]. In recent years, the rise of computational biology has led to the development of various web-based tools that predict TF–target gene relationships using unique algorithms and databases. The JASPAR database offers a collection of high-quality transcription factor DNA binding motifs, providing core data for bioinformatics analysis and a powerful predictive tool for experimental design [6,7]. Based on transcription factor knockout experiments, the KnockTF database systematically compiles data on the effects of knockouts on gene expression, providing valuable experimental evidence for TF function studies [8,9]. Here, we introduce a novel web application and package developed using the R programming language, designed to predict target genes of transcription factors and vice versa. Our application synergizes the predictive capacities of multiple web tools by intersecting their results to enhance reliability. Additionally, it incorporates gene expression correlation analysis as a filter to refine the predictions. This integrative approach offers a comprehensive and efficient strategy for the elucidation of transcriptional networks, providing a valuable resource for the research community to advance the field of gene regulatory mechanisms.

## 2. Materials and Methods

### 2.1. Transcription Factor Databases

This Shiny application has incorporated seven major TF-Target online tools (Table 1), namely hTFtarget [10], KnockTF [8,9], TRRUST [11,12], ENCODE [13], CHEA [14], GTRD [15], and ChIP_Atlas [16]. Additionally, this app incorporates the predicting results of TFs binding sites within upstream 2000 bps of the transcription start sites (TSS2000) of all genes using the Find Individual Motif Occurrences (FIMO) of MEME Suite (version 5.5.5) [17] and Position Weight Matrix Enrichment Analysis (PWMEnrich, version 4.38.0) [18] algorithms and the motif sequences of TFs obtained from the JASPAR database [7]. The TSS2000 sequences were downloaded from the UCSC website (https://hgdownload.cse.ucsc.edu/goldenpath/hg38/bigZips/upstream2000.fa.gz, accessed on 13 April 2024), and all parameters were set as default. All datasets were uploaded to the MySQL database.

### 2.2. Gene Expression Databases

To explore the correlation between TFs and target gene expression, this Shiny app integrates analyses using The Genotype-Tissue Expression (GTEx), The Cancer Genome Atlas Program (TCGA), and Cancer Cell Line Encyclopedia (CCLE) databases. The following is their basic introduction:The GTEx (Genotype-Tissue Expression) project database provides extensive reference data for gene expression and its variability in different normal human tissues, enabling researchers to understand how gene expression is influenced by genetic background on a broader scale [19].The TCGA (The Cancer Genome Atlas) database collects multidimensional cancer genomic data, including gene expression, mutations, copy number variations, and epigenetic data, offering strong support for the discovery of cancer biomarkers and therapeutic targets [20].The CCLE (Cancer Cell Line Encyclopedia) database provides a wealth of gene expression, mutation, and epigenetic characteristic data for numerous cancer cell lines, serving as a crucial resource for studying cell line-specific responses and drug screening [21].

We used Pearson’s method to analyze the correlation between the TFs contained in the R package and the expression of all genes and uploaded the results to the MySQL database. These resources provide extensive transcriptomic data from healthy and cancerous tissues or cells, essential for predicting TF–target interactions. We apply the correlation analysis between the expression of TFs and targets based on these datasets to enhance the predictive property. The analyses are performed using R, and results are dynamically presented in the app, offering users a clear view of the TF–gene expression relationships.

### 2.3. Software

Our Shiny application was exclusively developed using R (version 4.3.0) and encompasses all stages of development, from data extraction and correlation analysis to data visualization and user interface (UI) design. The following is the link to this Shiny application: https://jingle.shinyapps.io/TF_Target_Finder/ (accessed on 3 May 2024). The key R packages employed in the construction of this application are summarized in the table below, indicating their specific uses, such as UI construction, web data retrieval, and XENA database (https://xena.ucsc.edu/, accessed on 3 March 2024) [22] data extraction, MySQL database access, and visualization (Table 2).

### 2.4. Package

To facilitate researchers in utilizing this application more conveniently, I encapsulated the relevant functions within the application into an R package called “TFTF”, which has been uploaded to GitHub (https://github.com/WangJin93/TFTF, accessed on 3 May 2024). Once users install the “TFTF” package, they can access most of the prediction and visualization functionalities of this app within the R software environment. Additionally, users can run this app locally using the TFTF_app() function.

## 3. Results

### 3.1. Module 1: Procedures for the Prediction of the Target Genes of TF

This systematic approach enables a thorough analysis of TF–target gene interactions, bolstering the robustness of predictions by harnessing the combined strength of multiple dataset intersections. The procedural steps are as follows:(1)Input a TF name: Users initiated the process by selecting a TF of interest from a dropdown list.(2)Select TF-Target datasets: Then, users could the TF-Target datasets to include in the analysis. Notably, if “KnockTF” was selected, an additional interface element appeared: a slider for setting the log2 fold-change (Log2FC) threshold. This specification is necessary as KnockTF predictions are based on differential gene expression data following TF knockdown or knockout.(3)Select correlation datasets (optional): Users can further enhance the accuracy of predictions based on the correlation analysis results of TF and target gene expression in different tissues based on the TCGA and GTEx databases. Also, the threshold for the correlation coefficient is set to 0.3.(4)Results: By clicking the “Go” button, users commenced the predictive analysis. After a short wait, the prediction outcomes were displayed on the “All results” tab, which encompassed individual tool results and the intersected findings.(5)Intersection selection: Given that some tools may yield sparse predictions or lack data, we provided an option box to select well-predicted datasets for intersection analysis, which is visualized through a Venn diagram.(6)Visualize intersections: The “Venn diagram” tab allowed for the visualization of overlapping predictions across multiple tools using Venn or petal diagrams.(7)Individual dataset review: The “Individual dataset” tab enabled viewing and downloading detailed information for each tool’s predictive results.

Herein, we demonstrate the predictive workflow and outcomes using STAT3 as an example within this module. The integration analysis for prediction was conducted using 3 TF-Target datasets, viz. hTFtarget, CHIP_Atlas, and KnockTF, and 2 correlation analysis results, viz. cor_TCGA and cor_GTEx. For KnockTF, a log2 fold change (log2FC) threshold of 0.5 was set (Figure 1A). The results indicated that we identified 12735, 12383, 9008, 14116, and 10951 putative target genes from the five datasets, respectively (Appendix A), while the intersection of these tools yielded 1262 target genes (Figure 1B). Furthermore, the prediction results from each tool can be further viewed and downloaded in the “Individual dataset” tab (Figure 1C).

### 3.2. Module 2: Procedures for the Prediction of Upstream TFs of Target Genes

This integrated approach, combining gene expression correlation analysis with multi-dataset intersection, was designed to ensure a comprehensive and reliable prediction of TF–target gene interactions. The operational steps are detailed below:(1)Input a target gene symbol: GAPDH is an example of this.(2)Select datasets: Participants then chose the predictive tools to include in the analysis. Notably, if “KnockTF” was selected, an additional interface element appeared: a slider for setting the log2 fold-change (Log2FC) threshold, along with a checkbox to include only downregulated genes. This specification is necessary as KnockTF predictions are based on differential gene expression data following TF knockdown or knockout.(3)Select correlation datasets (optional): Users can further enhance the accuracy of predictions based on the correlation analysis results of TF and target gene expression in different tissues based on the TCGA and GTEx databases. Also, the threshold for the correlation coefficient is set to 0.3.(4)Results: Similar to Module 1, the predicted results were displayed on the “All results” Table. Also, users can select datasets with robust predictions for intersection analysis.

Using GAPDH as the exemplar, we incorporated five tools—hTFtarget, ENCODE, ChIP_Atlas, GTRD, and KnockTF—alongside correlation analysis with TCGA lung adenocarcinoma and GTEx lung tissue data (correlation coefficient threshold: 0.3) (Figure 2A). The combined prediction analysis of these seven datasets yielded 227, 137, 433, 591, 41, 1091, and 811 transcription factors, respectively (Appendix A). Finally, only three intersecting transcription factors, viz. STAT1, YY1, and FOXM1 were identified across these seven datasets (Figure 2B). In addition, upon removing GTRD and ENCODE from the “Select datasets to get intersection” dropdown, an intersection of the remaining five datasets revealed two more TFs: HIF1A and SMARCA4 (Figure 2C). Similarly, individual dataset results can be viewed and downloaded from the Individual dataset section (Figure 2D).

### 3.3. Module 3: Pan-Tissue Correlation Analysis between the Expression of Predicted TF–Target Pair

In this module, we utilized data from three publicly available databases to analyze the expression correlation of TF–target pairs across various tissue types. The integration of these analyses enables a comprehensive assessment of the expression relationship between the TFs and their potential target genes in a context-specific manner. The methodological steps are detailed as follows:(1)TF and target gene input: The user begins by selecting a transcription factor and entering the symbol for the target gene.(2)Database selection: The database(s) for analysis is/are chosen from among TCGA, GTEx, and CCLE. Notably, upon selecting TCGA, a popup menu appears, offering the user the option to include tumor data exclusively.(3)Correlation analysis parameters: Parameters for correlation analysis are set, including the selection of the analysis method and the establishment of thresholds for the correlation coefficient and *p*-value.(4)Correlation results and scatter plot: Data retrieval and correlation analysis are initiated by clicking the “Go” button. Subsequently, the results of the correlation analysis are presented, along with a scatter plot illustrating the expression correlation.(5)Plotting parameter: Options are provided to adjust parameters relevant to the scatter plot visualization.(6)Detailed scatter plot: Clicking on a row within the results table prompts a popup window that displays a detailed scatter plot for the expression of the two genes within a single tissue type.

We evaluated the correlation between FOXM1 and GAPDH across pan-cancer samples in the TCGA database using a Pearson correlation method with a coefficient threshold of 0.3 and a *p*-value threshold of 0.05 (Figure 3A). The results indicated a significant positive correlation between FOXM1 and GAPDH expression in the majority of cancers (Figure 3A,B, Appendix A). Upon selecting LUAD, we obtained the interface as shown in Figure 3C, where the scatter plot demonstrates the correlation between these two genes in TCGA-LUAD (correlation coefficient = 0.676). Also, significantly positive correlations between GAPDH and FOXM1 expression were also revealed in multiple tissue types based on the GTEx and CCLE databases (Figure 3D,E).

### 3.4. Module 4: TF–Target Regulation Network Analysis

The module was designed to predict the target genes of transcription factors (TFs) of interest based on gene differential expression analysis results uploaded by the user, utilizing multiple TF prediction databases, and to visualize the regulatory network. This module thus facilitates the elucidation of potential regulatory relationships by integrating user data with established TF prediction resources, supporting the discovery of novel insights into gene regulatory networks. The steps for utilizing this module are as follows:(1)Data upload: Users upload their gene expression differential analysis results. It is important to ensure that the column names in the uploaded file are consistent with those in the example data provided.(2)Differential gene selection criteria: Set the thresholds for selecting differentially expressed genes, specifically the log2 fold change (log2FC) and *p*-value.(3)TF different expression result: The ‘TF result’ page will exhibit the differential analysis results of TFs extracted from the user’s uploaded data.(4)Datasets selection: Choose the predictive tools to be included in the analysis for identifying TF–target gene relationships.(5)TF List Update: Upon input completion, the ‘TF to analysis’ input field automatically updates with a list of TFs. This list is generated based on the intersection of differentially expressed genes from the uploaded results and the TFs contained within the chosen predictive tools.(6)Network Visualization: Clicking the ‘Go’ button starts the predictive analysis process. After the analysis is complete, a network diagram is generated. Note that some TFs may not display target genes in the network diagram if no target genes are identified after intersecting the results from multiple tools. In such cases, it may be beneficial to reduce the number of tools included in the analysis to obtain more extensive information.(7)Plotting Data Interface: The ‘Plotting data’ interface will present the predicted results for TF–target genes.

Taking the differential analysis results from the GSE17025 dataset as an instance (Appendix A), we uploaded these results and set the thresholds for differentially expressed genes at log2FC = 1 and *p* value < 0.05. The “TF results” tab then displays 196 different expressed TFs extracted from the uploaded data (Figure 4A). In this analysis, we predicted TF target genes based on the hTFtarget and KnockTF datasets. The “TF to analysis” dropdown was automatically updated with TFs included in these two databases, and we selected all for analysis. The network visualization resulting from the intersection of the predicted outcomes and differentially expressed genes is illustrated in Figure 4B. The predictive outcomes and data for plotting can be viewed and downloaded in the Plotting data section (Figure 4C).

## 4. Discussion

In the application developed in this study, we crafted a suite of modular tools aimed at accurately predicting the interactions between transcription factors (TFs) and their target genes while taking into consideration the strengths and limitations of existing bioinformatic resources. Among the primary predictive resources, hTFtarget is known for its rich information on TF binding sites supported by high-throughput ChIP-seq data, yet its predictive results are constrained by specific experimental conditions and cell types [10]. The ENCODE project, with its foundation of extensive experimentally validated data, provides solid evidence for TF–target gene associations, though the universal applicability of this data can sometimes be limited [13,30]. Cistrome, by integrating a variety of bioinformatics tools and a vast collection of ChIP-seq datasets, offers more comprehensive predictions, but its accuracy may still be affected by the frequency of data updates and methodologies of data processing [31]. KnockTF, relying on data from gene knockout experiments, intuitively demonstrates TF functionality but often fails to adequately reflect the diversity of TFs across different biological settings [8,9]. Another study shares a similar purpose with our tool, providing TF–target interactions derived through these strategies, accompanied by confidence scores, as a resource for enhancing the prediction of TF activities [32]. Unfortunately, I was unable to access the resource link provided by the authors. However, it is evident that our tool offers more comprehensive functionalities, including bidirectional prediction of TF–target regulatory relationships, prediction of TF–target regulatory networks based on user data, and correlation analysis of candidate TF–target relationships across multiple tissues.

Module 1 of our application enhances the precision and reliability of target gene predictions by merging data from various databases, using the intersection of datasets to fill gaps that reliance on a single database might miss. Module 2 improves the accuracy of upstream TF predictions by combining gene expression correlation analysis with dataset integration, ensuring the biological relevance of the results. Module 3, through pan-tissue correlation analysis, empowers researchers to evaluate the expression relationships between TFs and their potential target genes in specific biological contexts. Biological networks elucidate the intricate interplay between various molecular entities within cells, interactions that are crucial in determining cellular behavior [4]. At the initial stage of constructing transcription regulatory networks, it is imperative to establish accurate associations between transcription factors (TFs) and their target genes, which involves identifying their regulatory activities and determining whether they function as activators or repressors. Advanced experimental techniques like chromatin immunoprecipitation (ChIP), in conjunction with high-throughput sequencing technologies, enable us to identify TF binding sites on a genome-wide scale. This capability lays the foundation for revealing and constructing sophisticated transcription regulatory networks [33,34]. Finally, Module 4 of this app fuses user data with established TF prediction resources to analyze the TF–target gene regulatory network, offering new perspectives for exploring gene regulatory networks.

Based on the analysis results from Modules 2 and 3 of this app, we identified FOXM1 as a potential upstream transcription factor regulating GAPDH expression. The regulatory mechanism between the two has been confirmed in our previous studies. We found that GAPDH, commonly used as a reference gene, is significantly upregulated in most cancers and holds significant clinical relevance. Further mechanistic studies revealed its upregulation to be under the transcriptional regulation of FOXM1 [35]. This confirms the utility and reliability of this application and R package.

## 5. Conclusions

In conclusion, this application not only effectively utilizes the advantages of existing database resources but its modular design also enhances the comprehensiveness and precision of target gene predictions, which is crucial for revealing the role of TFs in diverse biological contexts. This integrated and systematic approach will greatly strengthen the ability of researchers to understand and explore the complexities of gene regulatory networks. However, it is important to note that the current version of the application only supports the prediction of human TF–target gene interactions and has not yet been expanded to include other species. This limitation restricts its application in broader biological research and cross-species comparative analyses. With future enrichment of databases and refinement of algorithms, we anticipate that the application will gradually extend support to the transcription regulatory network analysis of more species, thereby deepening our understanding of complex biological networks.

## Figures and Tables

**Figure 1 biomolecules-14-00749-f001:**
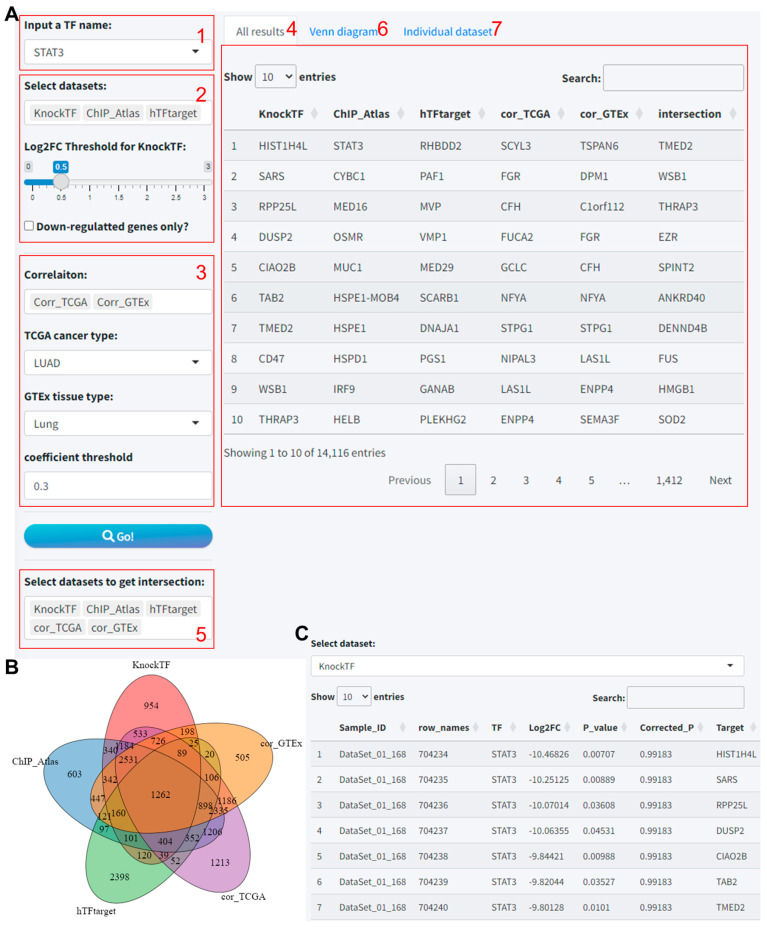
Visualization of STAT3 target prediction results utilizing the TF-Target Finder Shiny app. (**A**) Parameter setting and display of predictive outcomes; (**B**) a Venn diagram depicting the intersections of predicted results from five datasets; (**C**) the Individual dataset interface showing results from a single dataset, exemplified by KnockTF predictions. TF, transcription factor.

**Figure 2 biomolecules-14-00749-f002:**
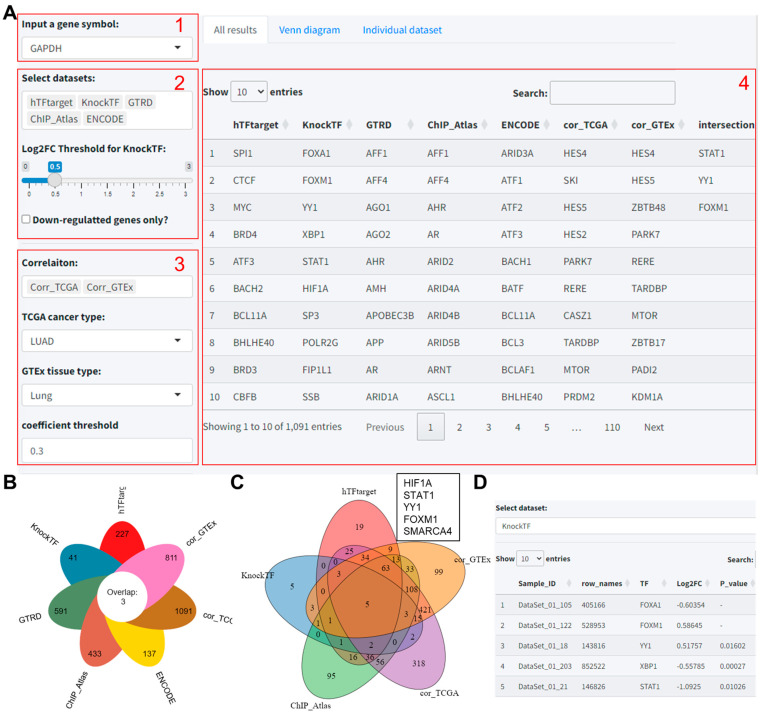
Predictive analysis of TFs regulating GAPDH using the TF-Target Finder Shiny app. (**A**) Parameter setting and display of predictive results; (**B**) a flower plot illustrating the outcomes of five predictive tools and two correlation analyses, along with their intersections; (**C**) a Venn diagram showing the intersections of predictions from three tools and two correlation analyses; (**D**) the Individual dataset interface with KnockTF predictions exemplifying results from a single dataset. TF, transcription factor.

**Figure 3 biomolecules-14-00749-f003:**
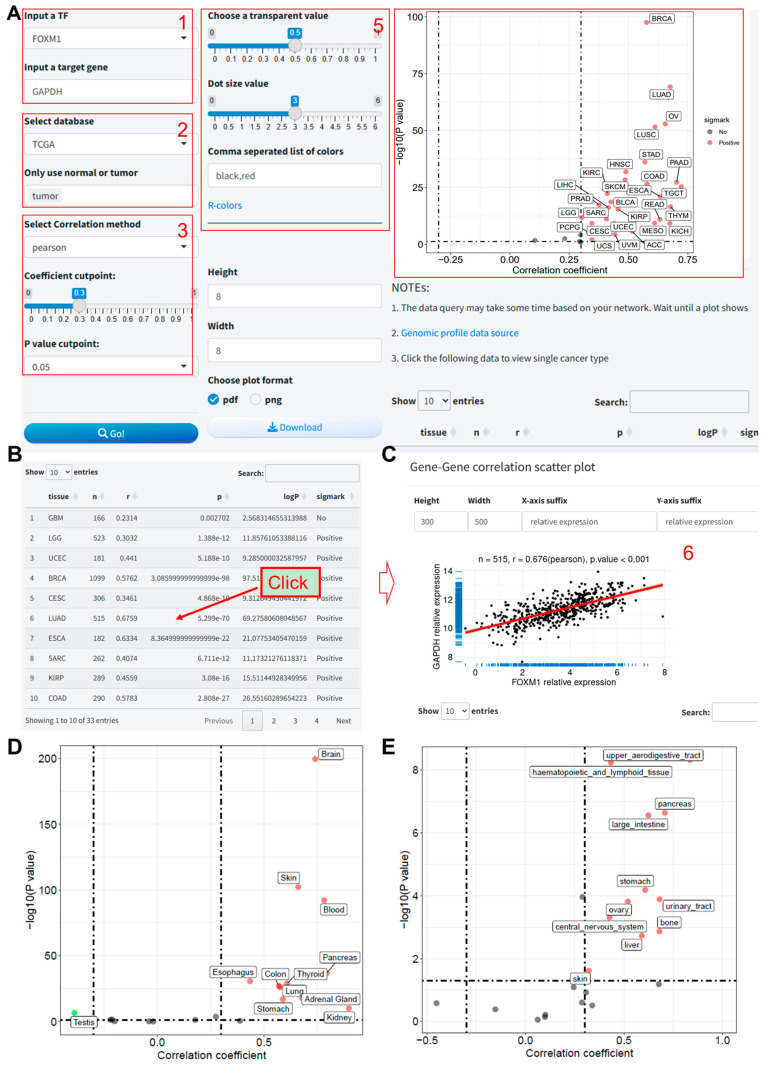
Correlation analysis of FOXM1 and GAPDH expression in TCGA pan-cancer samples using the TF-Target Finder Shiny app. (**A**,**B**) Parameter setting and display of predictive results; (**C**) scatter plot illustrating the correlation between GAPDH and FOXM1 in the TCGA-LUAD dataset. Scatter plots showing the results of correlation analysis based on GTEx (**D**) and CCLE (**E**) databases. TF, transcription factor. TCGA, The Cancer Genome Atlas. LUAD, lung adenocarcinoma. GTEx: Genotype-Tissue Expression. CCLE, Cancer Cell Line Encyclopedia.

**Figure 4 biomolecules-14-00749-f004:**
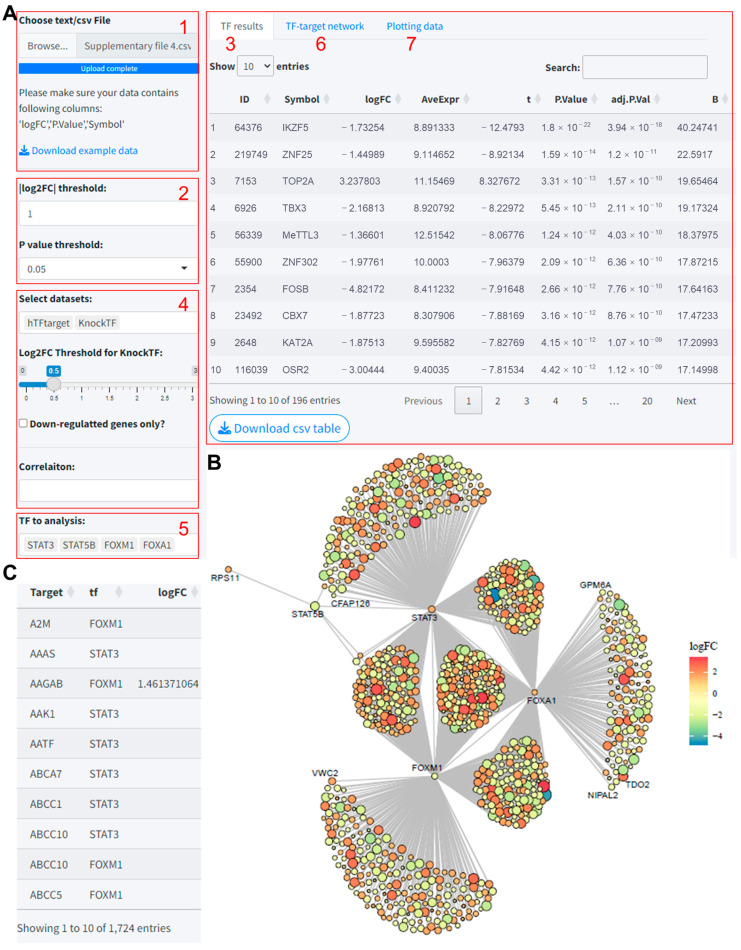
TF–targets regulation network analysis based on differential gene expression analysis results using the TF-Target Finder Shiny app. (**A**) Parameter setting and display of differential expression results for TFs extracted from uploaded data; (**B**) visualization of the TF–targets regulation network; (**C**) viewing and downloading prediction results in Plotting data Table TF, transcription factor.

**Table 1 biomolecules-14-00749-t001:** Information of databases used in this shiny application.

Data Type	Datasets	DB Link	Data Source	Evidence
TF database	hTFtarget [10]	https://guolab.wchscu.cn/hTFtarget/#!/ (accessed on 1 March 2024)	http://bioinfo.life.hust.edu.cn/hTFtarget#! (accessed on 1 March 2024)	ChIP-Seq data
	KnockTF [8,9]	https://bio.liclab.net/KnockTFv1/ (accessed on 1 March 2024)	https://bio.liclab.net/KnockTF/index.php (accessed on 1 March 2024)	Knockdown/knockout
	TRRUST [11,12]	https://www.grnpedia.org/trrust/ (accessed on 1 March 2024)	https://www.grnpedia.org/trrust/ (accessed on 1 March 2024)	Pubmed
	ENCODE [13]	https://www.encodeproject.org/ (accessed on 1 March 2024)	https://maayanlab.cloud/Harmonizome/dataset/ENCODE+Transcription+Factor+Targets (accessed on 1 March 2024)	ChIP-Seq data
	CHEA [14]	https://maayanlab.cloud/chea3/ (accessed on 1 March 2024)	https://maayanlab.cloud/static/hdfs/harmonizome/data/cheappi/gene_attribute_edges.txt.gz (accessed on 1 March 2024)	ChIP-seq data, Co-expression
	GTRD [15]	https://gtrd.biouml.org/ (accessed on 5 March 2024)	https://gtrd20-06.biouml.org/bioumlweb/# (accessed on 5 March 2024)	ChIP-Seq data
	ChIP_Atlas [16]	https://chip-atlas.org/ (accessed on 8 March 2024)	https://chip-atlas.dbcls.jp/data/hg38/target/STAT3.1.tsv (accessed on 8 March 2024)	ChIP-Seq data
	JASPAR [7]	https://jaspar.elixir.no/ (accessed on 5 March 2024)	https://jaspar.elixir.no/downloads/ (accessed on 5 March 2024)	motifs
Gene expression database	GTEx [19]	https://www.genome.gov/Funded-Programs-Projects/Genotype-Tissue-Expression-Project (accessed on 2 March 2024)	https://xenabrowser.net/datapages/?dataset=gtex_rsem_isoform_tpm&host=https%3A%2F%2Ftoil.xenahubs.net (accessed on 2 March 2024)	gene expression RNAseq
	TCGA [20]	https://portal.gdc.cancer.gov/ (accessed on 3 April 2024)	https://xenabrowser.net/datapages/?dataset=tcga_RSEM_gene_tpm&host=https%3A%2F%2Ftoil.xenahubs.net (accessed on 3 April 2024)	gene expression RNAseq
	CCLE [21]	https://sites.broadinstitute.org/ccle/ (accessed on 3 April 2024)	https://xenabrowser.net/datapages/?dataset=ccle%2FCCLE_DepMap_18Q2_RNAseq_RPKM_20180502&host=https%3A%2F%2Fucscpublic.xenahubs.net (accessed on 3 April 2024)	gene expression RNAseq

**Table 2 biomolecules-14-00749-t002:** Key R packages and their functions in shiny application development.

R Package	Version	Function
shiny [23]	1.8.0	Building interactive web application UI
bs4Dash [24]	2.3.0	Advanced UI design with Bootstrap 4 integration
jsonlite [25]	1.8.8	Parsing JSON data
UCSCXenaShiny [26]	1.1.10	Extracting gene expression data from XENA database
ggplot2 [27]	3.4.4	Data visualizations
igraph [28]	1.6.0	Network graph visualization
VennDiagram [29]	1.7.3	Venn diagram visualization

## Data Availability

The data presented in this study derived from the following resources available in the public domain: hTFtarget [10], KnockTF [8,9], TRRUST [11,12], ENCODE [13], CHEA [14], GTRD [15], and ChIP_Atlas [16], which were downloaded from their respective websites. The URLs for these resources are listed in Table 1. The transcription factor target gene data based on FIMO and PWMEnrich will be made available by the authors on request.

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
