# Peer review of "TFTF: An R-Based Integrative Tool for Decoding Human Transcription Factor–Target Interactions"

_biomolecules, 2024, doi:10.3390/biom14070749_

Round 1

Reviewer 1 Report (Previous Reviewer 2)

Comments and Suggestions for Authors

The manuscript was improved, for specialized readers focused on the area it is a very helpful article, however for a reader unfamiliar with the topics it is difficult to understand. I consider that it will be of great help and usefulness to the area of ​​molecular biology.

Author Response

Dear Professor,

Thank you for your thoughtful and constructive feedback. I am pleased to hear that you have found this manuscript helpful and beneficial to professional readers in the field of molecular biology. Thank you for your recognition of the improvements made. This manuscript is designed to provide tools for researchers in need, so it focuses on providing detailed technical information. However, I will continue to improve the R package /shinyapp in order to make it more accessible to a wider audience based on user feedback. Your insights will be invaluable in guiding these efforts.

Reviewer 2 Report (Previous Reviewer 3)

Comments and Suggestions for Authors

The authors have resolved the majority of issues that I identified in
my first review. The fact that the source code is now available on
github and available for anyone is in my eyes improving the ms
greatly as it allows for people to identify potential issues. In
addition with this actual application that is modular in design and relies
on databases  it also opens up for the research community to add more
features to the application. There is currently no license file added
to the git repository and I would encourage the authors to add a
suitable open source license as a final step making the code base
accessible for anyone.

I still think there are room for improvement when it comes to
describing how a query is actually handled. Are all queries that goes
to external databases happening in real-time for every instance of the
application or is the MySQL databases used as a backend pre-populated
with data when possible, and in that case when exactly is this
done. Along the same line, I am still missing details on how the
dropdowns in some of the views are created. I presume the list that is
found in the TF list tab of the application is the backbone of the dropdown and if I only select a subset of databases only these will be possible to find in the dropdown, but this is not fully clear.

Overall there are still some explanations to terms etc that are missing. For example:

- Full information on what the different tissues in the correlation
  plots corresponds to. It could be in the form of a popup window in
  the plot or even somewhere else easy to find.
- DataSet_01_168 and similar terms found in the TF targets prediction
  table, what is this name exactly referring to? Is there any better
  naming that can be used to make the actual name more informative?

I would encourage the author to go over the application and highlight names/headers etc that are not common knowledge and make sure to add a clarification to these terms to make the tool more useful for anyone that are not that used to working with these datasets and might not have a rich vocabulary connected to these kind of databases.

The shiny application is largely using the same layout and design as the previous version and are to large extant showing similar issues.
Queries are sometimes not finishinig (perhaps they are too extensive and
eventually the query just stops). Other times empty results takes long
time and should be possible to report with some index that do not
require time to actual perform searches or at least try to tell the user that a given value might be missing from the database.

Here is an example of challenging searches that never finished even after several attempts.

Target -> TFs:

RNU6-880P only hfTarget, KnockTF
Takes a bit of time even though result list is empty

RNU6-880P only hfTarget, KnockTF, FIMO_JASPAR
Times out after around 5 minutes, no results obtained

RNU6-880P only hfTarget, KnockTF, PWMEnrich_JASPAR
Times out after around 5 minutes, no results obtained

The long waiting time often makes the browser session disconnect to
the server making it impossible to know if the query is just complex
and takes a long time or if there are actual issues with the obtaining results.

Author Response

Comments 1: The authors have resolved the majority of issues that I identified in my first review. The fact that the source code is now available on github and available for anyone is in my eyes improving the ms greatly as it allows for people to identify potential issues. In addition with this actual application that is modular in design and relies on databases  it also opens up for the research community to add more features to the application. There is currently no license file added to the git repository and I would encourage the authors to add a suitable open source license as a final step making the code base accessible for anyone.

Response 1: Thank you for your valuable feedback and for bringing this important aspect to our attention. We have now added an appropriate open-source license to our Git repository to ensure that everyone can access and use the codebase legally.

We chose the MIT License as it aligns well with our project's goals and provides sufficient freedom for users. The LICENSE file has been added to the root directory of the repository.

We appreciate your encouragement to make our work more accessible and are grateful for your insightful suggestions.

Comments 2: I still think there are room for improvement when it comes to describing how a query is actually handled. Are all queries that goes to external databases happening in real-time for every instance of the application or is the MySQL databases used as a backend pre-populated with data when possible, and in that case when exactly is this done. Along the same line, I am still missing details on how the dropdowns in some of the views are created. I presume the list that is found in the TF list tab of the application is the backbone of the dropdown and if I only select a subset of databases only these will be possible to find in the dropdown, but this is not fully clear.

Response 2: Thank you for your question. Regarding the data query method, based on the feedback from the previous submission's reviewer about the limitations of web scrapers due to network and target website stability, I have addressed this in the updated version. I downloaded publicly available complete databases from these online tools and uploaded them to a MySQL database. The updated R package uses APIs to retrieve data from the MySQL database in real-time for integration and analysis. In the TF→Target module, the TF dropdown list includes all TFs from the R package database. The dataset checkboxes automatically update based on the selected TF, displaying only datasets that contain the selected TF. This modification was completed before resubmitting the manuscript. Similarly, based on your suggestion, I have further refined the shiny app. In the Target→TF module, I adopted a similar strategy where the datasets are automatically updated based on the input target gene, thus enhancing the user experience and reducing unnecessary wait times.

 Overall there are still some explanations to terms etc that are missing. For example:

Comments 3:  - Full information on what the different tissues in the correlation   plots corresponds to. It could be in the form of a popup window in   the plot or even somewhere else easy to find.

Response 3: I appreciate your suggestions. Regarding the tissue abbreviations in the correlation analysis scatter plots, such as BLCA and GBM, these are official abbreviations from the TCGA project. I have now optimized the code to include a column displaying the full names of these abbreviations to enhance user experience.

Comments: 4 - DataSet_01_168 and similar terms found in the TF targets prediction   table, what is this name exactly referring to? Is there any better   naming that can be used to make the actual name more informative?

 Response 4In fact, the identifier "DataSet_01_168" refers to a dataset ID from the KnockTF database. In the Individual Dataset interface, I strive to provide users with comprehensive information sourced from the original analysis websites, facilitating further filtering and analysis of results. To clarify this, I have added an explanatory note in the interface to inform users of the data's origin: "The information in the table, aside from TF and Target, represents specific details obtained from the corresponding prediction tools."

Comments 5: I would encourage the author to go over the application and highlight names/headers etc that are not common knowledge and make sure to add a clarification to these terms to make the tool more useful for anyone that are not that used to working with these datasets and might not have a rich vocabulary connected to these kind of databases.

 Response 5Thank you for this important suggestion. I have made several optimizations to improve user instructions and clarify terms that might not be familiar to all users. I will continue to gather feedback and make further improvements to ensure the tool is user-friendly and accessible to researchers with varying levels of familiarity with these datasets.

Comments 6: The shiny application is largely using the same layout and design as the previous version and are to large extant showing similar issues.

 Queries are sometimes not finishinig (perhaps they are too extensive and eventually the query just stops). Other times empty results takes long time and should be possible to report with some index that do not require time to actual perform searches or at least try to tell the user that a given value might be missing from the database.

 Response 6Thank you for your feedback on this matter. The issue of empty results is related to the dropdown list settings previously mentioned. As explained, the dataset dropdown list now updates in real-time based on the user-selected TF or Target, showing only relevant datasets, which significantly reduces invalid data retrievals.

Regarding the long analysis times and server disconnections, these issues arise because the shinyapp is currently deployed on shinyapp.io with limited access due to the lack of a VIP subscription, leading to traffic and memory constraints. To address this, I have created an R package that includes the app's code, allowing users to run the shiny app locally via RStudio using the TFTF_app function. Users can also directly use the R package's functions for data retrieval and visualization within the R environment.

Comments 7: Here is an example of challenging searches that never finished even after several attempts.

 Target -> TFs:

 RNU6-880P only hfTarget, KnockTF

 Takes a bit of time even though result list is empty

 RNU6-880P only hfTarget, KnockTF, FIMO_JASPAR

 Times out after around 5 minutes, no results obtained

 RNU6-880P only hfTarget, KnockTF, PWMEnrich_JASPAR

 Times out after around 5 minutes, no results obtained

 The long waiting time often makes the browser session disconnect to

 the server making it impossible to know if the query is just complex

 and takes a long time or if there are actual issues with the obtaining results.

 Response 7Thank you for providing this example. In the optimized app, the Target dropdown list now includes only genes present in these datasets. The gene you mentioned, RNU6-880P, is not present in any of the datasets, which was an oversight when constructing the gene list by including all genes from the Ensembl database. This has now been corrected. As previously explained, the dataset dropdown list has been optimized to prevent such issues.

Thank you again for your invaluable feedback. Your suggestions have greatly contributed to enhancing the functionality and user experience of the tool. Please do not hesitate to let me know if you have any further suggestions or comments.

This manuscript is a resubmission of an earlier submission. The following is a list of the peer review reports and author responses from that submission.

Round 1

Reviewer 1 Report

Comments and Suggestions for Authors

The author presented a new webtool to anayze multiple data sources related to transcription factors and target genes. 

the author presented easy webtool with 4 modules.

1. Even if the author present one example per module, they did not evaluate if the findings identified well-known association and novel ones. The author needs to evaluate their tool and show the pertinence of their findings.

2.In the module 1, the description of the example does not match with the figures. The author needs to correct this.

3. The author did not compare their tool with other tools in this same idea and did not use benchmark to evaluate it. The author may want to look at this paper https://www.ncbi.nlm.nih.gov/pmc/articles/PMC6673718/ and other papers can be found

Comments on the Quality of English Language

English language in the paper is fine. No issues detected. However the sections of website and the documentation is in chinese.

The author needs to provide all documents in English even on their website to help international users.

Reviewer 2 Report

Comments and Suggestions for Authors

Interesting manuscript, however still confusing in the explanations of the transcription factor interaction predictions, it is suggested to review it wid.

Reviewer 3 Report

Comments and Suggestions for Authors

1 Review of TF-Target Finder: An R…
═══════════════════════════════════

  The well written manuscript with the fairly long and winding title
  "TF-Target Finder: An R Web Application Bridging Multiple Predictive
  Models for Decoding Transcription Factor-Target Interactions" presents
  an easy to use web-tool that make it possible to combine information from several
  large databases. It will both help in identify interesting
  transcription factors in the human genome, but can also present data
  on gene expression patterns associated with these. The presentation of
  the tool in this manuscript is overall clear and easy to follow, but
  it there are many key factors not described in detail and it is also
  too narrow in scope.

1.1 Major concerns
──────────────────

  • I can not find any link to the source code for building the
    application. This type of manuscripts need to share the source code
    so that the research community can have a chance of understanding
    how data is extracted and combined. This is both to avoid adding
    more "black boxes" to research that really should be as reproducible
    as possible. In addition it will also make it possible to discover
    and remove bugs in the code. The paper is citing a R package named
    UCSCXenaShiny which is an excellent example of how to proceed. Not
    only is the source code available, but they have also made an
    r-package of the code making it easy to deploy and expand upon. This
    would also allow researchers to add other public or private datasets
    to be queried with this application.
  • I lack details on how data is extracted from different databases. In
    addition it is not clear exactly what data are stored in the local
    MySQL database and when this database is populated. Is it a one-time
    event or is it querying the remote databases and populating the
    local database at launch? What info is actually in this database?
  • The fact that rvest is used to scrape data using web-pages
    potentially makes the application sensitive to alterations on
    external web-pages and might brake this tool. Have this been
    considered in designing the application.
  • The application could be much more useful if it were possible to
    query in a programming environment using an API (once again for an
    example see the UCSCXenaShiny library).
  • Searches with transcription factors giving 0 hits should preferably
    be possible to filter out and handle differently instead of the
    users having to wait for queries that yield no output. 6 out 10 of
    the first listed TF in the application does yields no data. Not sure
    how the list of available TF is created, but if this is created from
    the databases I am not sure why so many empty data sets are created.
  • There needs to be a better explanation to what one should upload to
    the "TF-target net". To just state that the file should contain a
    set of columns with specific names is not enough. What values should
    be find in the columns etc.

1.2 Minor concerns
──────────────────

  • Title should be shorter and also clarify that only human data are
    available
  • Row 61: What MySQL database is this sentence referring to see a also
    comment under major concerns
  • A bit too often the application renders an errors. For example "An
    error has occurred. Check your logs or contact the app author for
    clarification." It would be good if this was minimized in terms of
    how often it happens and when it does try to give more useful
    feedback. This seems to happen when the transcription factor is not
    present? For example, trying the INTS11 and GAPDH on the "Pan-tissue
    cor" yields errors.
  • Row 101: How is this drop down list created?
  • Row 122: This lists four tools, but not encode, even though encode
    is shown in figure 1. The text describing what is seen in figure 1
    is not what is actually shown in figure 1.
  • Row 301: The described modular design is a perfect reason to open up
    to source code and make it easy for user to add other data sources.
  • The reference list is at places a bit thin. Reference 3 is an
    example from prokaryotes which is not wrong, but could perhaps also
    have reference to something more directly tied to human. Reference
    21 to the UCSCXenaShiny library is included, but not references to
    the other R libraries mentioned in the paper is this by choice and
    in that case why?
  • Is there any reason not to use harmonize for more of the data sets?
    The harmonize database contains also other data sets that might be
    useful in context of this application. Have you consider making more
    of these available in the application.
  • Links to external web resources in the application should preferably
    open a new tab/page instead of opening them in the same page that
    app is running as shiny applications tend to lose its current state
    and reload the application as a new instance even if one go back in
    the browser.